# Decision Trees for Glaucoma Screening Based on the Asymmetry of the Retinal Nerve Fiber Layer in Optical Coherence Tomography

**DOI:** 10.3390/s22134842

**Published:** 2022-06-27

**Authors:** Rafael Berenguer-Vidal, Rafael Verdú-Monedero, Juan Morales-Sánchez, Inmaculada Sellés-Navarro, Oleksandr Kovalyk, José-Luis Sancho-Gómez

**Affiliations:** 1Departamento de Ciencias Politécnicas, Universidad Católica de Murcia UCAM, 30107 Guadalupe, Spain; rberenguer@ucam.edu; 2Departamento de Tecnologías de la Información y Comunicaciones, Universidad Politécnica de Cartagena, 30202 Cartagena, Spain; juan.morales@upct.es (J.M.-S.); oleksandr.kovalyk@edu.upct.es (O.K.); josel.sancho@upct.es (J.-L.S.-G.); 3Hospital General Universitario Reina Sofía, 30003 Murcia, Spain; inmasell@um.es

**Keywords:** optical coherence tomography (OCT), peripapillary OCT, retinal nerve fiber layer (RNFL), RNFL thickness asymmetry, retinal imaging analysis, glaucoma, decision trees

## Abstract

**Purpose**: The aim of this study was to analyze the relevance of asymmetry features between both eyes of the same patient for glaucoma screening using optical coherence tomography. **Methods**: Spectral-domain optical coherence tomography was used to estimate the thickness of the peripapillary retinal nerve fiber layer in both eyes of the patients in the study. These measurements were collected in a dataset from healthy and glaucoma patients. Several metrics for asymmetry in the retinal nerve fiber layer thickness between the two eyes were then proposed. These metrics were evaluated using the dataset by performing a statistical analysis to assess their significance as relevant features in the diagnosis of glaucoma. Finally, the usefulness of these asymmetry features was demonstrated by designing supervised machine learning models that can be used for the early diagnosis of glaucoma. **Results**: Machine learning models were designed and optimized, specifically decision trees, based on the values of proposed asymmetry metrics. The use of these models on the dataset provided good classification of the patients (accuracy 88%, sensitivity 70%, specificity 93% and precision 75%). **Conclusions**: The obtained machine learning models based on retinal nerve fiber layer asymmetry are simple but effective methods which offer a good trade-off in classification of patients and simplicity. The fast binary classification relies on a few asymmetry values of the retinal nerve fiber layer thickness, allowing their use in the daily clinical practice for glaucoma screening.

## 1. Introduction

Glaucoma is an eye disease which damages the optic nerve head (ONH) and can produce complete loss of vision. After cataracts, it is the leading cause of irreversible blindness worldwide [1]. In 2013 there were 64.3 million people aged 40–80 with glaucoma worldwide, and this number is estimated to increase to 111.8 million in 2040 [2]. The term glaucoma describes ocular disorders with multi-factorial etiology which usually are united with clinical intraocular-pressure-associated optic neuropathy. There are different types of glaucoma, classically divided into the categories of primary or secondary open-angle or angle-closure glaucoma [3]. Primary open-angle glaucoma (POAG) is the most common form of glaucoma, where the drainage angle formed by the cornea and iris remains open, but the trabecular meshwork is partially blocked, causing an increase in the intraocular pressure (IOP) [4]. Secondary forms of glaucoma are caused by various ocular or systemic diseases; most of them are associated with high IOP, but not all, such as normal-tension glaucoma [5]. Primary open-angle glaucoma is usually associated with an increase in the intraocular pressure which modifies the appearance of the optic nerve, causing the neuroretinal rim of the optic nerve to become progressively thinner, thereby enlarging the optic-nerve cup. The different types of glaucoma have in common a slow progressive degeneration of retinal ganglion cells and their axons, resulting in a distinct appearance of the optic disc and a concomitant pattern of visual loss [6]. Patients with glaucoma typically lose peripheral vision and may lose all vision if not treated properly [7].

Glaucoma is a silent and progressive illness which is initially asymptomatic. The diagnosis is based on tonometry to assess the intraocular pressure, campimetry to evaluate the visual field, and imaging modalities to analyze the morphology of the ONH and its retinal layers. The internal structure of the eye can be imaged by retinal fundus images and optical coherence tomography [8].

Optical coherence tomography (OCT) is an imaging modality used in the assessment of the glaucomatous damage [9]. The OCT provides images with backscatter produced by the differences in the refractive index between adjacent retinal tissues [10,11]. The initial time-domain (TD)-OCT was surpassed by spectral domain (SD)-OCT, which offers increased axial resolution and faster scanning speeds [12]. SD-OCT allows one to analyze and quantify eye parameters, such as the thickness of the peripapillary XXXXX retinal nerve fiber layer (RNFL) for monitoring glaucoma [13,14], and evaluate neurostructural and vascular changes (see, e.g., [15]).

The damage produced by glaucoma alters the morphology of ocular structures, such as the RNFL and ONH, which allows for discrimination between glaucomic and healthy eyes. Those structural changes in the ONH or the thinning of the RNFL result in differences between the eyes of the same individual. For this reason, asymmetry has been analyzed and considered as an early indicator of glaucoma [16] using, e.g., the difference in values of intraocular pressure [17], central corneal thickness [18], corneal hysteresis [19], neuroretinal rim width [20] or as in this work, RNFL thickness [21]. While some studies determine the limits of the normal interocular asymmetry in retinal layers considering thickness measurements with OCT in normal subjects (see, e.g., [22,23,24]), other studies use RNFL thickness measurements to evaluate the diagnostic capabilities of intereye and intraeye differences to identify early primary open-angle glaucoma [25] or propose an early glaucoma discriminator index [26].

In this work, we focus on primary open-angle glaucoma and propose a set of asymmetry metrics based on the thickness of the RNFL of both eyes. Statistical analysis was performed to quantify the importance of each metric as a relevant characteristic for glaucoma diagnosis. A simple classifier was then applied to the most discriminant metric, obtaining a decision tree for the classification of healthy and glaucoma patients. This model offers a simple but effective tool for the screening and early detection of glaucoma that can be used in the daily clinical practice, similarly to the inferior-superior-nasal-temporal (ISNT) rule [27], the cup-to-disc ratio (CDR) [28] and the rim-to-disc ratio (RDR) [29] measurements. Note that these previous indicators are obtained from retinal fundus images, instead of OCT, which is one of the novelties of this work.

After this introduction, the rest of the article is organized as follows: Section 2 provides all the details of the dataset used and the proposed metric to evaluate the asymmetry. Section 3 shows the statistical analysis of the asymmetry metrics and describes the classification models, along with the metrics providing the greatest discrimination between healthy and glaucoma patients. Finally, Section 4 closes the article with a discussion and the conclusions.

## 2. Materials and Methods

This section describes the image acquisition procedure and the methods used in the assessment of the diagnosis of the disease.

### 2.1. Image Acquisition Procedure

We took a set of 2D peripapillary B-scan optical coherence tomography (OCT) images centered at the optic nerve head as raw material, from both healthy individuals and unhealthy patients with glaucoma. The OCT image dataset used in this work was acquired by the Ophthalmology Service of the Hospital General Universitario Reina Sofía (HGURS, Murcia, Spain) using a Spectralis OCT S2610-CB (Heidelberg Engineering GmbH, Heidelberg, Germany) from October 2018 to November 2020 from 287 individuals. These images were anonymized according to the criteria of the Human Ethics Committee.

The patients included in the study were divided into two groups. In the first group, *G1*, patients diagnosed with simple chronic glaucoma recruited in the *Glaucoma Section* of the HGURS; in the second group, *G2*, were primary care patients without any type of ocular pathology (verified by ophthalmologists) that could influence the morphology of the optic nerve. An exclusion criterion was the presence of opacities in the transparent media (corneal alterations and advanced cataracts) that prevented obtaining an assessable OCT. Another reason for excluding a patient was that the alteration in the confidence indices of the visual field persisted in two consecutive tests. The patients of G2 had IOP equal to or below 22 mmHg, and no lesions suggestive of glaucomatous neuropathy were detected. The sample size of patients was calculated by taking into account the population of *Area VII* of the city of Murcia and the prevalence of simple chronic glaucoma (3.5% in the population older than 40 years), while applying a confidence level of 95% and a 3% margin of error.

The spectral-domain optical coherence tomography (SD-OCT), also called Fourier-domain OCT (FD-OCT), was acquired by means of an 870 nm wavelength super-luminescent diode (SLD), scanning a cylindrical section of the retinal layers centered on the optic disc. This cylindrical section, also called a B-scan, was projected from polar to Cartesian coordinates resulting in a image with a resolution of 768 × 496 pixels, a bit depth of 8 bits/pixel in grayscale and a *z*-scaling of 3.87 µm/pixel, as illustrated in Figure 1. In order to facilitate the analysis, the cylindrical B-scan is normally divided into a group of 6 sectors whose names refer to their positions, namely, temporal (T), temporal superior (TS), nasal superior (NS), nasal (N), nasal inferior (NI) and temporal inferior (TI). Note that sectors (T) and (N) use an angle of $90^{{}^{\circ}}$, whereas the rest of sectors use an angle of $45^{{}^{\circ}}$. As shown in Figure 2, the naming of the sectors is symmetrical with respect to the right and left eyes. Thus, the analysis of the asymmetry can be performed by simple comparison of the thickness measurements between each paired eye sector. The relations between polar coordinates measured in degrees and Cartesian coordinates measured in pixels is provided in Table 1 and outlined at the bottom of Figure 1.

As described in [30], expert ophthalmologists diagnosed both eyes of all individuals in the database using three possible levels of glaucoma disease: healthy, not healthy and uncertain. Given the aim of this work to compare the asymmetry between the eyes of healthy and glaucoma patients, only individuals with two healthy eyes (160 individuals) and with both eyes with glaucoma disease (47 individuals) were selected (see Table 2). Hence, patients with only one eye diagnosed with glaucoma or those for whom the expert diagnosis was inconclusive were not considered.

Note that the sizes of the two groups are not comparable; i.e., the number of healthy individuals was larger than the number of unhealthy ones. This may suggest biased results in a subsequent classification process. Nevertheless, it is important to highlight that the individuals in the database were collected directly from patients attending the HGURS ophthalmology service; thus, the distribution of the two sample groups is similar to what can be found in a real clinical setting for glaucoma diagnosis.

### 2.2. RNFL Segmentation and Thickness Calculation

The next step is devoted to the segmentation of the images in order to estimate the thickness of the retinal nerve fiber layer (RNFL). The segmentation of the RNFL under analysis poses a great challenge due to the characteristics of the images, and more specifically, the features of the retinal layer under analysis. As described in [9,10], speckle noise, a low level of contrast and irregularly shaped morphological features are frequently present in OCT datasets. In addition, since the RNFL is the innermost layer of the retina, artifacts due to the shadowing effect of the retinal veins appear in the images, increasing the difficulty of precisely defining the layers in specific sectors of the eye. The most commonly used approaches for this segmentation are comprehensively reviewed in [30], and some of them are relevant to the dataset used here.

The segmentation method chosen for this work is the algorithm provided by Spectralis software version 6.9.4.0. The thickness of the RNFL is calculated from the positions of the upper and lower boundary of the RNFL provided by the segmentation algorithm, scaled from pixels to µm according to the OCT resolution, as shown in Figure 3. Unlike other methods where the thickness value at each angle of the B-scan can be obtained, the Spectralis device provides only the average thickness of the RNFL for each sector of the eye (T, TS, NS, N, NI and TI) and the thickness average value of the global circumpapillary contour (G). Therefore, these mean section thickness values were used for the inter-eye asymmetry study (Section 3).

### 2.3. Proposals for Asymmetry Metrics

In this section, a set of inter-eye asymmetry measures of RNFL thickness are proposed. The difference between the mean value of the RNFL thickness in the right eye and its corresponding thickness in the left eye is the base operation for all measures. Some metrics calculate absolute differences, in order to analyze only the difference between eyes by isolating the effect of which particular eye shows greater thickness. Nevertheless, the influence of the sign of the difference in the RNFL thickness on the discrimination between healthy patients and glaucoma patients has also been explored with other metrics. Additionally, given that the thickness of the retinal layers varies between individuals, a set of normalizers have been proposed to provide relative asymmetry values, with the objective of decoupling to some extent the specific dimensions of the eyes of the patients.

The first attempt to measure the asymmetry was with the following difference
(1)$$\delta_{S,i}=w_{S,i}^{r}-w_{S,i}^{l},$$
where *S* denotes the specific sector (T, TS, NS, N, NI, TI or G) over which the asymmetry is calculated; *i* indicates the patient number; and $w_{S,i}^{r}$ and $w_{S,i}^{l}$ refer to the mean thickness of the RNFL layer in the sector *S* calculated for the right and left eyes of patient *i*, respectively. By applying the modulus to the previous equation, the direction in which the subtraction is calculated becomes irrelevant. Thus, the following measure is obtained:(2)$$|\delta_{S,i}\left|=\right|w_{S,i}^{r}-w_{S,i}^{l}|.$$

The following proposals for the measurement of the asymmetry normalize previous measures with the sum of the RNFL thickness of both eyes in the corresponding sector *S* of the patient *i*:(3)$$\Delta_{S,i}=\frac{\delta_{S,i}}{w_{S,i}^{r}+w_{S,i}^{l}}=\frac{w_{S,i}^{r}-w_{S,i}^{l}}{w_{S,i}^{r}+w_{S,i}^{l}},$$
and
(4)$$|\Delta_{S,i}|=\frac{|w_{S,i}^{r}-w_{S,i}^{l}|}{w_{S,i}^{r}+w_{S,i}^{l}}.$$

The normalization can also be performed with the average thickness in each sector of all the patients in the dataset:(5)$${\bar{\Delta }}_{S,i}=\frac{\delta_{S,i}}{w_{S}^{r}+w_{S}^{l}}=\frac{w_{S,i}^{r}-w_{S,i}^{l}}{w_{S}^{r}+w_{S}^{l}},$$
and
(6)$$|{\bar{\Delta }}_{S,i}|=\frac{|w_{S,i}^{r}-w_{S,i}^{l}|}{w_{S}^{r}+w_{S}^{l}},$$
where the values $w_{S}^{r}$ and $w_{S}^{l}$ can be obtained as
(7)$$w_{S}^{r}=\frac{1}{N}\sum_{i=1}^{N}w_{S,i}^{r},$$
and
(8)$$w_{S}^{l}=\frac{1}{N}\sum_{i=1}^{N}w_{S,i}^{l},$$
i.e., the mean value of the RNFL thickness for each sector of both eyes. The values obtained with the dataset of OCTs used in this work (N=160+47=207) are gathered in Table 3.

Finally, the last proposed normalizations divide the thickness difference by the sum of the average thickness of the complete RNFL in both eyes of patient *i*:(9)$${\bar{\bar{\Delta }}}_{S,i}=\frac{\delta_{S,i}}{w_{G,i}^{r}+w_{G,i}^{l}}=\frac{w_{S,i}^{r}-w_{S,i}^{l}}{w_{G,i}^{r}+w_{G,i}^{l}},$$
and
(10)$$|{\bar{\bar{\Delta }}}_{S,i}|=\frac{|w_{S,i}^{r}-w_{S,i}^{l}|}{w_{G,i}^{r}+w_{G,i}^{l}},$$
where $w_{G,i}^{r}$ and $w_{G,i}^{l}$ are, respectively, the average thickness of the complete RNFL in the right eye and in the left eye of patient *i*.

## 3. Results

This section firstly addresses the statistical characterization of the RNFL thickness values using all previous asymmetry metrics. The mean, standard deviation and *p*-value were calculated for the subsets of healthy and glaucoma patients. Next, the chosen machine learning models, decision trees, are briefly described, and the design process for glaucoma screening is detailed.

### 3.1. Statistical Characterization of the Asymmetry Metrics

The thickness asymmetry has been calculated with the proposed metrics for each sector of the OCT and for the overall RNFL, considering two subsets, healthy and glaucoma patients (see Table 2). In order to analyze the relevance of each asymmetry metric, statistical characterization has been performed. Table 4 gathers the mean value, standard deviation and *p*-value of each metric for all sectors (TS, T, TI, NS, N and NI) and for entire OCT scan (G) for the two subsets of healthy and glaucoma patients. The values of the non-normalized metrics $\delta_{S}$ and $|\delta_{S}|$ are given in µm, whereas the values of the normalized metrics are dimensionless and range from −1 to 1 in the case of $\Delta_{S}$, ${\bar{\Delta }}_{S}$ and ${\bar{\bar{\Delta }}}_{S}$, and from 0 up to 1 in the case of the metrics based on the absolute value, i.e., $|\Delta_{S}|$, $|{\bar{\Delta }}_{S}|$ and $|{\bar{\bar{\Delta }}}_{S}|$.

As can be seen in Table 4, the metrics not using absolute value ($\delta_{S}$, $\Delta_{S}$, ${\bar{\Delta }}_{S}$ and ${\bar{\bar{\Delta }}}_{S}$) have mean values centered at zero and standard deviations lower in the group of healthy individuals than in the group of glaucoma patients. According to the definitions of these asymmetry metrics $\delta_{S}$, $\Delta_{S}$, ${\bar{\Delta }}_{S}$ and ${\bar{\bar{\Delta }}}_{S}$, the statistical distributions of such asymmetry values will have be normal when the number of patients is high enough, due to the central limit theorem. The means of these normal distributions will always be approximately zero, considering that the direction in which the subtraction is calculated has been arbitrarily chosen, and the asymmetry must occur equiprobably in both directions (from the left eye to the right, or vice versa). Then, since the values follow a normal distribution with a zero mean, the variance of the distributions will be the distinguishing factor in the screening of healthy and glaucoma patients.

On the other hand, metrics based on the absolute value ($|\delta_{S}|$, $|\Delta_{S}|$, $|{\bar{\Delta }}_{S}|$ and $|{\bar{\bar{\Delta }}}_{S}|$) have mean values greater than zero and standard deviations for the glaucoma subset slightly higher than for the subset of healthy patients. The absolute values of the asymmetry metrics cause normal distributions to become half-normal distributions [31]. The mean of the half-normal distribution is
(11)$$\mu =\sigma \sqrt{\frac{2}{\pi }},$$
where σ is the standard deviation of the initial normal distribution. That is, the mean of the new half-normal distribution is determined, proportionally, by the variance of the initial normal distribution, allowing the use of thresholds for the discrimination of healthy and glaucoma patients. Note that a greater variance will imply a higher mean value after taking the absolute value of the asymmetry for a specific sector.

The third characteristic gathered in Table 4 is the *p*-value, which determines the statistical significance of the hypothesis that the asymmetry of the RNFL thickness is statistically related to the occurrence of glaucoma disease. Then, a low *p*-value indicates that the asymmetry metric in a given sector has sufficient statistical significance in the occurrence of glaucoma between the healthy and glaucoma subset of patients. Taking this into account, since the *p*-values of the metrics $|\Delta_{S}|$ and $|{\bar{\bar{\Delta }}}_{S}|$ are the smallest of all the proposed metrics, $|\Delta_{S}|$ and $|{\bar{\bar{\Delta }}}_{S}|$ are the most useful and relevant as discriminant variables in patient classification.

With the aim of easing the understanding of the distribution of the values provides by each asymmetry metric, Figure 4 shows notched box plots of the proposed RNFL thickness asymmetry metrics for the healthy (**h**) and glaucoma (**g**) subsets of patients, depicted for each eye sector (TS, T, TI, NS, N and NI) and for the overall global thickness value (G). For each box, the central mark indicates the median, and the bottom and top edges of the box indicate the 25th and 75th percentiles, respectively. These percentiles delimit the so-called interquartile range. The whiskers extend to the most extreme values not considered outliers. Outliers are values located more than 1.5 times the interquartile range outside the upper or lower boundary of the box. The tapered and shaded regions, called notches, display the variability of the median between values. The width of a notch is computed so that boxes whose notches do not overlap have different medians at the 5% significance level—i.e., the true medians do differ with 95% confidence.

As can be seen in Figure 4, these graphs corroborate the insights concluded above. The values of metrics $\delta_{S}$, $\Delta_{S}$, ${\bar{\Delta }}_{S}$ and ${\bar{\bar{\Delta }}}_{S}$ are distributed around zero, and there is greater dispersion for the subset of glaucoma patients. The box plots of $|\delta_{S}|$, $|\Delta_{S}|$, $|{\bar{\Delta }}_{S}|$ and $|{\bar{\bar{\Delta }}}_{S}|$ depict how the variances of the distributions not using absolute values have been transformed into the corresponding means of the related metrics with absolute values, according to Equation (Equation 11). In addition, it can also be seen qualitatively how the notches with the least overlap for healthy and glaucoma patients are those corresponding to metrics $|\Delta_{S}|$ and $|{\bar{\bar{\Delta }}}_{S}|$, especially in sectors TS and TI, although the distribution of asymmetry values considering the entire layer, G, has the greater separation. As will be proven numerically later, this fact graphically justifies the use of these metrics as the characteristics for the design of the classifier.

### 3.2. Decision Trees

Artificial intelligence (AI) is increasingly being incorporated into the diagnostic process in healthcare due to its ability to analyze data with complex artificial networks and to learn automatically, especially through machine learning (ML) and deep learning (DL) [32,33]. In this work, instead of using complex classification methods based on machine learning to diagnose glaucoma (see, e.g., [34,35,36]), we chose classification trees [37] for their simplicity, with the aim of transferring them to daily clinical practice in glaucoma screening.

Classification tree denotes the *classification and regression tree* methodology (CART) used to describe decision tree algorithms that are used for classification learning tasks. A decision tree is a supervised machine learning algorithm with a tree-like structure which repeatedly splits the input dataset into classes, taking into account one exploratory variable at a time. These trees are used when the target variable is categorical and can assume only one of two mutually exclusive values (healthy and glaucoma, in our case) [38,39].

Considering the RNFL thickness in all sectors (TS, T, TI, NS, N and NI) and the global mean (G) of the peripapillary OCT, the proposed asymmetry metrics have been applied to generate a vector with the features of the subsets of healthy and glaucoma patients. With these features as input variables, a fitted binary classification decision tree has been designed for each asymmetry metric taking into account the labeling of each class (healthy or glaucoma patient).

The designed binary trees split branching nodes based on the values of the input features. One of the parameters of the model is the depth of the tree, which is related to the model complexity, and therefore, to the computational cost. The classification trees have been developed using five folds in the cross-validation. The results of the classification trees have been analyzed controlling the maximum depth of the trees (or maximum number of splits), ξ. Initially, the weights of the inputs corresponding to the class healthy and to the class glaucoma, $w_{h}$ and $w_{g}$, respectively, were set to unity, i.e., $w_{h}=1.0$ and $w_{g}=1.0$. Table 5 gathers the classification loss for observations not used for training in deep decision trees (ξ=15) and shallower trees (ξ=3). The largest value of ξ for deep decision trees depends on the number of input features used, which in our case, were the seven asymmetry measurements performed in the six sectors and in the global average of the RNFL thickness. As can be seen, the classification loss decreases as the maximum number of splits is reduced, providing the best values for the metrics |Δ|, $\bar{\bar{\Delta }}$ and $|\bar{\bar{\Delta }}|$. The best values have been highlighted in bold in Table 5, and their corresponding decision trees and confusion matrices are shown in Figure 5. The representation of each model is a binary tree where each root node represents a single input variable (asymmetry feature) and a split point on that variable. The leaf nodes of the tree contain an output variable which is used to predict if the input corresponds to a healthy or a glaucoma patient. Regarding the confusion matrices, they provide four outcomes:True positive (TP): glaucoma patient predicted as glaucoma (top left element),False positive (FP): healthy patient predicted as glaucoma (bottom left element),True negative (TN): healthy patient predicted as healthy (bottom right element),False negative (FN): glaucoma patient predicted as healthy (top right element).

In order to assess the performances of these three trees, the following parameters have been computed from the confusion matrix:Accuracy:
(12)$$ACC=\frac{TP+TN}{TP+TN+FP+FN}.$$Sensitivity (recall or true positive rate, TPR):
(13)$$TPR=\frac{TP}{TP+FN}.$$Specificity (true negative rate, TNR):
(14)$$TNR=\frac{TN}{TN+FP}.$$Precision (positive predictive value, PPV):
(15)$$PPV=\frac{TP}{TP+FP}.$$

As can be seen in Table 6, these decision trees provide high accuracy (all of them higher than 87%), high specificity (higher than 95%) and high precision (higher than 79%). In order to improve the results, i.e., increase the sensitivity while keeping high values for the remaining parameters, the weight of the inputs corresponding to the class glaucoma has been increased from 1.0 to 1.5. Table 7 contains the classification loss values of the new models considering this weight in the glaucoma input observations.

Based on the results of classification loss gathered in Table 7, the model which offers the best performance with the minimum number of splits is |Δ| with ξ=3. Figure 6 shows the resulting model and its corresponding confusion matrix. This classification tree provides the confusion matrix whose parameters are collected in Table 8.

Finally, in order to further increase the number of true glaucoma patients detected, i.e., the sensitivity, and considering that the asymmetry metric |Δ| offers the lowest classification loss and taking into account that the asymmetry values range from 0 to 1 (since this measure is normalized by the sum of the RNFL thickness of each eye in the corresponding sector), the asymmetry metric ${\left|\Delta \right|}^{0.5}$ is proposed. The nonlinear operation of applying the square root to the non-negative real values provided by the metric |Δ| decompresses the distribution of the values corresponding to healthy and glaucoma patients. This process of encoding with this decompressive power-law nonlinearity is called gamma compression [40]. As can be seen in Table 9, which gathers the statistical parameters of the metric ${\left|\Delta \right|}^{0.5}$, the mean values for healthy and glaucoma patients have increased in all sectors. In addition, the distances between the mean values for the subsets of healthy and glaucoma patients have further increased, compared to the values provided by |Δ|. This effect can be also observed in Figure 7, where the notched box plots of each subset are further apart. The resulting decision tree using ${\left|\Delta \right|}^{0.5}$ and the corresponding confusion matrixes are shown in Figure 8. The parameters of accuracy, sensitivity, specificity and precision are gathered in Table 10.

## 4. Discussion

This article has delved into the hypothesis that the asymmetry between the two eyes can be used as an indicator of glaucoma. For this purpose, new asymmetry metrics have been defined for the thickness of the RNFL obtained from OCTs of both eyes of the same patient. In this work, the mean thickness of each sector of the RNFL was provided by the segmentation of Spectralis; however, segmentation methods such as those described in [30,41] will be used in the future to address the relationship of the RNFL thickness at each angle of the B-scan in the diagnosis of glaucoma.

Through a statistical analysis of the RNFL thickness values, the asymmetry metrics that provide the greatest distinction between the values of the subset of healthy and glaucoma patients have been chosen. Using these metrics, supervised machine learning models have been designed—decision trees. The resulting models are classifiers that partitioning the feature space of the asymmetry values in the sectors of the RNFL and building binary decision trees recursively can create meaningful outputs that predict being healthy or suffering from glaucoma.

The proposed decision trees are simple but effective models which provide a binary classification decision (healthy or glaucoma) by splitting branching nodes based on the input asymmetry features. Their simplicity or reduced complexity is one of their main advantages. They allow for rapid classification of new observations, since it is much simpler to evaluate just one or two logical conditions than to compute scores using complex nonlinear equations for each group. For example, in [42] complex machine learning classifiers (conditional inference trees, logistic model tree, C5.0 decision tree, random forest and extreme gradient boosting *XGBoost*) use OCT parameters to diagnose glaucoma, achieving slightly better performances (average accuracy 0.8818, average sensitivity 0.9166, average specificity 0.8507 and average area under the curve 0.9459) than decision trees. However, these models can be seen as black boxes which produce results based solely on the input data using an algorithm, which prevents clinicians from understanding how variables are being combined to make such predictions. On the contrary, decision trees proposed in this paper allow one to study and analyze the importance of the input asymmetry features, and ophthalmologists may obtain clinical insight from the explanations. In addition, the visual diagram of a decision tree provides a simple explanation for the reason for classifying a patient as healthy or glaucomic. In daily clinical practice it is much easier to explain if–then statements than complex nonlinear equations.

Another advantage of decision trees is their implicit feature selection, since the top few nodes on which the tree is split are the most important variables within the set. In all the trees designed, the characteristic considered in the root is the asymmetry in the global (G) thickness of the RNFL, followed by the asymmetry in the TI sector.

The decision trees have some limitations, such as their tendency to overfit when very complex, in which case they generally have low bias. In this work, this drawback has been overcome by restricting the maximum number of splits. The best trees proposed for the diagnosis of glaucoma have, at most, up to three divisions without degrading the screening performance. Experimentally, it has been found that three partitions suppose an adequate trade-off between simplicity and performance, allowing, if necessary, for the possibility of being applied by hand, without the support of a computer.

Another limitation of decision trees is that they can be unstable because small variations in the input data can generate a completely different tree. This situation is also solved by limiting the maximum number of splits, which improves the ability of the model to generalize to new input data. In addition, as future work, the size of the dataset will be increased (both healthy and patients with glaucoma, keeping the balance between classes) allowing us to check the performance of the designed trees.

One more future line of work is the analysis of the performances of complex machine learning models made to estimate the upper bound of the classification results that can be obtained using the proposed database and to study the differences between the results of the proposed decision trees and these complex models, even though they cannot be easily implemented by ophthalmologists in daily clinical practice.

In conclusion, decision tree models are easy to understand and implement (even by hand if necessary), which gives them a strong advantage when compared to other analytical models. The results obtained in this paper confirm that the proposed machine learning classifier, the decision tree, offers satisfactory performance and a high capacity for generalization, providing a simple model for glaucoma screening which can be used in daily clinical practice.

## Figures and Tables

**Figure 1 sensors-22-04842-f001:**
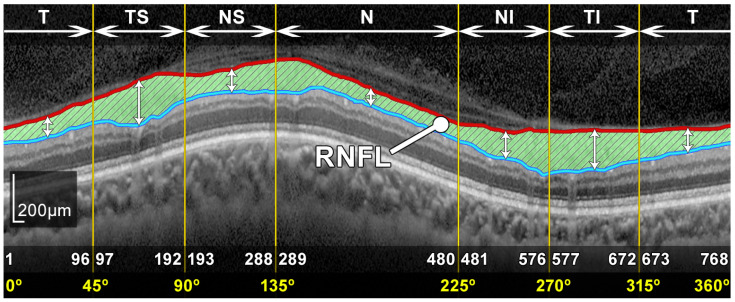
Segmented RNFL in a peripapillary B-scan OCT image. Relations of the sectors (T, TS, NS, N, NI and TI) measured in degrees and pixels.

**Figure 2 sensors-22-04842-f002:**
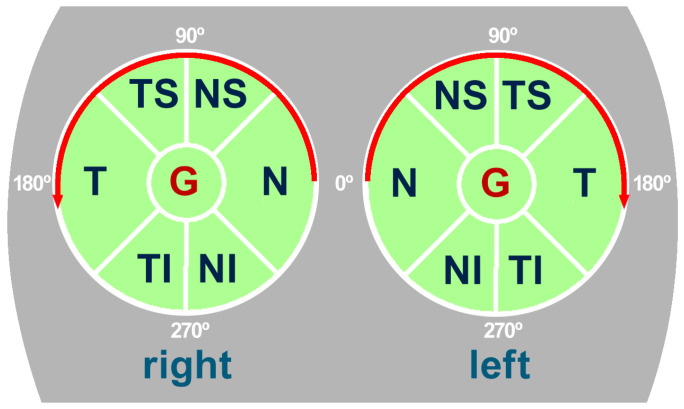
Diagram of the situation of the sectors of the eye in relation to the face of the subject.

**Figure 3 sensors-22-04842-f003:**
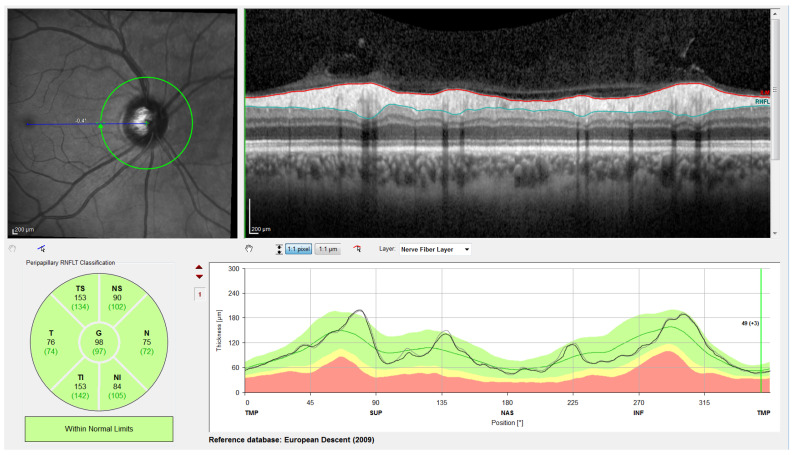
Screenshot of Spectralis. From left to right and top to bottom: Retinal fundus image centered on the optic disc (the green circle indicates the location of the peripapillary B-scan, which is shown on the right with the segmentation of the RNFL); 2D peripapillary B-scan OCT with Cartesian coordinates; estimated mean values for RNFL layer thickness for each sector and the overall mean; rectified outline of the RNFL with estimated thickness and reference values according to the database European Descent (2009).

**Figure 4 sensors-22-04842-f004:**
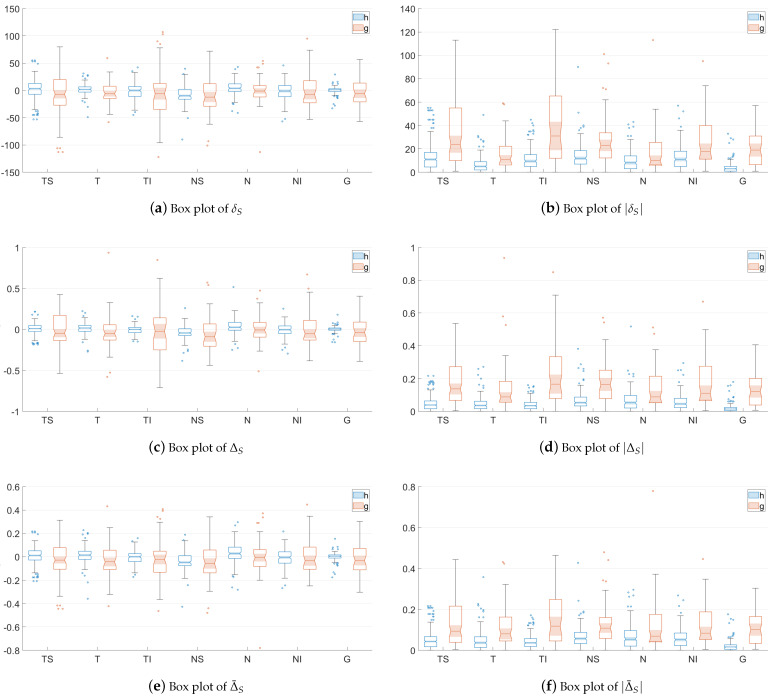
Notched box plots of the proposed RNFL thickness asymmetry metrics for the healthy (h) and glaucoma (g) subsets of patients, depicted for each eye sector (TS, T, TI, NS, N and NI) and for the overall global thickness value (G).

**Figure 5 sensors-22-04842-f005:**
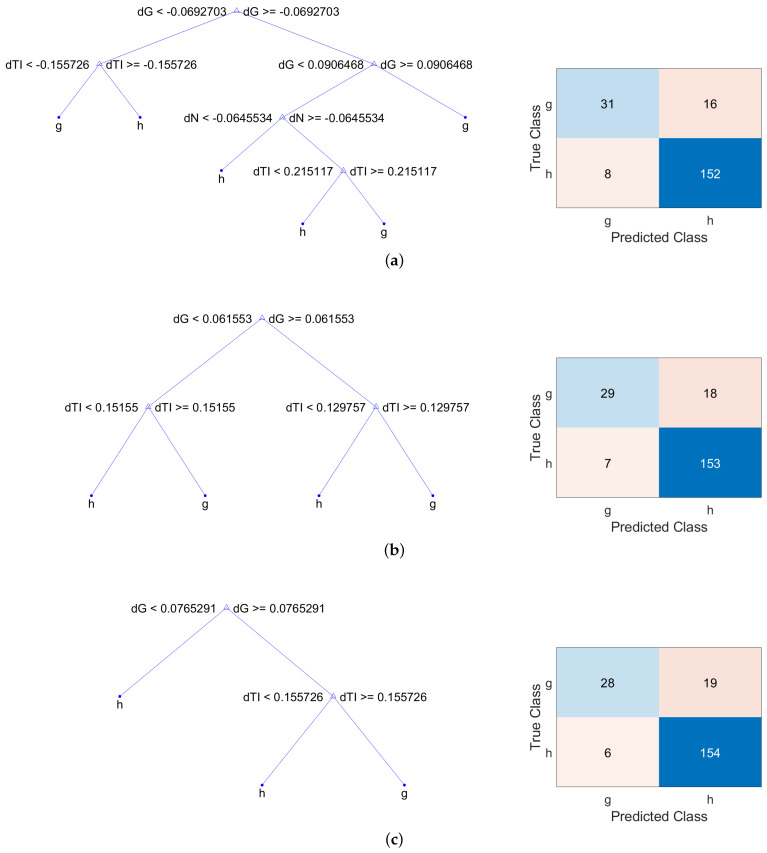
Decision trees and confusion matrices of the best results of Table 5. (**a**) Decision tree (**left**) and confusion matrix (**right**) for the metric $\bar{\bar{\Delta }}$, using $w_{h}=1$, $w_{g}=1$ and maximum number of splits ξ=5. (**b**) Decision tree (**left**) and confusion matrix (**right**) for the metric |Δ|, using $w_{h}=1$, $w_{g}=1$ and maximum number of splits ξ=3. (**c**) Decision tree (**left**) and confusion matrix (**right**) for the metric $|\bar{\bar{\Delta }}|$, using $w_{h}=1$, $w_{g}=1$ and maximum number of splits ξ=3.

**Figure 6 sensors-22-04842-f006:**
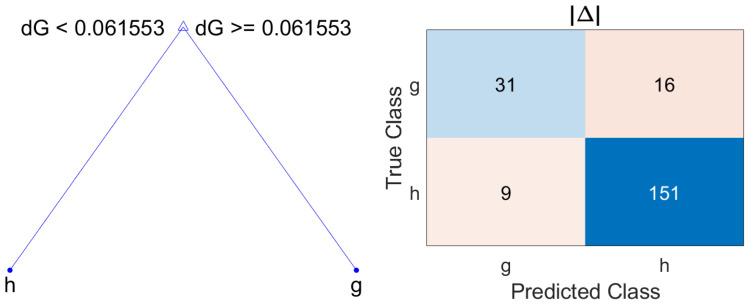
Decision tree and confusion matrix for metric |Δ|, using $w_{h}=1$, $w_{g}=1.5$ and ξ=3.

**Figure 7 sensors-22-04842-f007:**
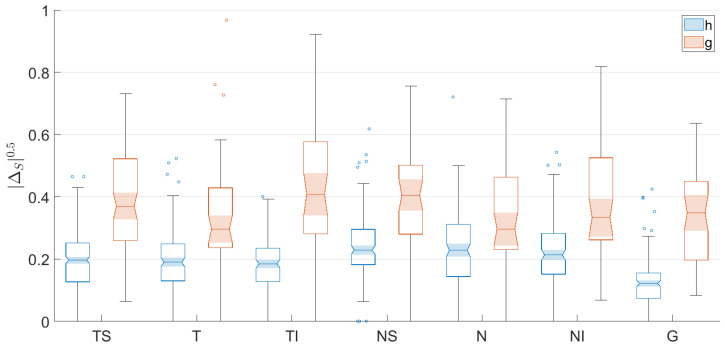
Notched box plots of the proposed RNFL thickness asymmetry metric $|\Delta_{S}{|}^{0.5}$ for the healthy (h) and glaucoma (g) subsets of patients, depicted for each eye sector (TS, T, TI, NS, N and NI) and for the overall global thickness value (G).

**Figure 8 sensors-22-04842-f008:**
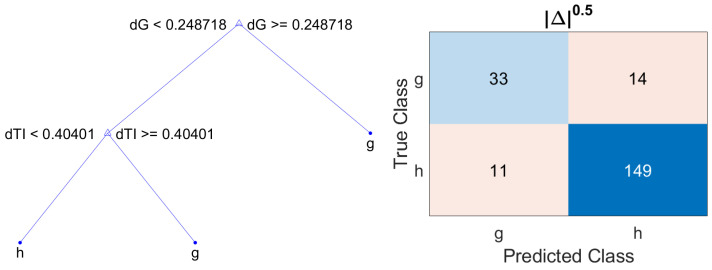
Decision tree and confusion matrix for metric ${\left|\Delta \right|}^{0.5}$, using $w_{h}=1$, $w_{g}=1.5$ and ξ=3.

**Table 1 sensors-22-04842-t001:** Relations between the polar and Cartesian coordinates in the OCT projection.

		Temporal (T)	Temporal Sup. (TS)	Nasal Sup. (NS)	Nasal (N)	Nasal Inf. (NI)	Temporal Inf. (TI)	Temporal (T)
Polar	Min	$0^{{}^{\circ}}$	$45^{{}^{\circ}}$	$90^{{}^{\circ}}$	$135^{{}^{\circ}}$	$225^{{}^{\circ}}$	$270^{{}^{\circ}}$	$315^{{}^{\circ}}$
(degrees)	Max	$45^{{}^{\circ}}$	$90^{{}^{\circ}}$	$135^{{}^{\circ}}$	$225^{{}^{\circ}}$	$270^{{}^{\circ}}$	$315^{{}^{\circ}}$	$360^{{}^{\circ}}$
Cartesian	Min	1	97	193	289	481	577	673
(pixels)	Max	96	192	288	480	576	672	768

**Table 2 sensors-22-04842-t002:** Summary of the OCT dataset used to evaluate the asymmetry metrics.

	Healthy (h)	Glaucoma (g)	Total
Age (years)	59.10±12.70	70.15±9.21	61.61±12.85
Gender (male/female)	53/107	19/28	72/135
Total	160 patients	47 patients	207 patients
	(320 OCTs)	(94 OCTs)	(414 OCTs)

**Table 3 sensors-22-04842-t003:** Mean values in µm of the RNFL thickness calculated for all sectors in both eyes. Values used in asymmetry metrics ${\bar{\Delta }}_{S,i}$ and $|{\bar{\Delta }}_{S,i}|$.

	TS	T	TI	NS	N	NI	G
Right	$w_{TS}^{r}$	$w_{T}^{r}$	$w_{TI}^{r}$	$w_{NS}^{r}$	$w_{N}^{r}$	$w_{NI}^{r}$	$w_{G}^{r}$
	128.1691	68.7826	130.0773	101.2029	74.1932	106.1353	93.9469
Left	$w_{TS}^{l}$	$w_{T}^{l}$	$w_{TI}^{l}$	$w_{NS}^{l}$	$w_{N}^{l}$	$w_{NI}^{l}$	$w_{G}^{l}$
	126.3527	68.0531	132.8454	109.5266	70.9469	106.6957	93.9710

**Table 4 sensors-22-04842-t004:** Statistical characterization of the asymmetry metrics proposed for RNFL thickness. Mean value, standard deviation and *p*-value of each metric for the two subsets of healthy and glaucoma patients.

**Sector**	$\delta_{S}$			$|\delta_{S}|$		
	**Healthy**	**Glaucoma**	***p*-Value**	**Healthy**	**Glaucoma**	***p*-Value**
TS	2.7563±16.8122	−1.3830±45.3539	$5.0247\times 10^{-1}$	12.7438±11.2637	34.1489±29.4522	$6.1489\times 10^{-6}$
T	2.1500±9.6793	−4.1064±20.1022	$3.3495\times 10^{-2}$	7.0125±6.9896	15.2979±13.4987	$2.6754\times 10^{-4}$
TI	−2.3375±14.9220	−4.2340±51.8594	$7.0975\times 10^{-1}$	11.7500±9.4463	40.0213±32.7291	$1.0413\times 10^{-6}$
NS	−7.5938±16.6878	−10.8085±34.9370	$8.4612\times 10^{-1}$	14.3938±11.3149	27.8298±23.4227	$4.0740\times 10^{-3}$
N	4.2875±12.5123	−0.2979±26.0576	$2.9617\times 10^{-1}$	10.0000±8.6271	17.3617±19.2641	$2.2671\times 10^{-2}$
NI	−1.0625±16.5372	1.1489±34.5989	$3.6061\times 10^{-1}$	12.8125±10.4606	26.7660±21.5973	$5.7953\times 10^{-5}$
G	0.6563±6.5895	−2.3404±25.8728	$5.5860\times 10^{-1}$	4.2188±5.0937	20.6383±15.4855	$4.5656\times 10^{-9}$
**Sector**	$\Delta_{S}$			$|\Delta_{S}|$		
	**Healthy**	**Glaucoma**	***p*-Value**	**Healthy**	**Glaucoma**	***p*-Value**
TS	0.0099±0.0619	−0.0040±0.2357	$6.4483\times 10^{-1}$	0.0470±0.0413	0.1824±0.1469	$6.7965\times 10^{-8}$
T	0.0142±0.0647	−0.0361±0.2235	$1.0267\times 10^{-1}$	0.0481±0.0453	0.1465±0.1714	$3.9634\times 10^{-4}$
TI	−0.0075±0.0531	−0.0170±0.3159	$7.7715\times 10^{-1}$	0.0416±0.0338	0.2396±0.2036	$4.9500\times 10^{-8}$
NS	−0.0359±0.0798	−0.0494±0.2155	$8.8592\times 10^{-1}$	0.0666±0.0566	0.1750±0.1329	$2.9386\times 10^{-5}$
N	0.0300±0.0876	0.0069±0.1854	$4.8085\times 10^{-1}$	0.0674±0.0634	0.1386±0.1217	$4.8106\times 10^{-4}$
NI	−0.0052±0.0772	0.0058±0.2342	$4.9736\times 10^{-1}$	0.0582±0.0508	0.1779±0.1502	$1.8119\times 10^{-6}$
G	0.0033±0.0346	−0.0162±0.1804	$5.4845\times 10^{-1}$	0.0214±0.0272	0.1428±0.1094	$9.2247\times 10^{-10}$
**Sector**	${\bar{\Delta }}_{S}$			$|{\bar{\Delta }}_{S}|$		
	**Healthy**	**Glaucoma**	***p*-Value**	**Healthy**	**Glaucoma**	***p*-Value**
TS	0.0108±0.0661	−0.0054±0.1782	$5.0240\times 10^{-1}$	0.0501±0.0443	0.1342±0.1157	$6.1539\times 10^{-6}$
T	0.0157±0.0707	−0.0300±0.1469	$3.3450\times 10^{-2}$	0.0512±0.0511	0.1118±0.0986	$2.6933\times 10^{-4}$
TI	−0.0089±0.0568	−0.0161±0.1972	$7.0988\times 10^{-1}$	0.0447±0.0359	0.1522±0.1245	$1.0371\times 10^{-6}$
NS	−0.0360±0.0792	−0.0513±0.1658	$8.4615\times 10^{-1}$	0.0683±0.0537	0.1321±0.1112	$4.0816\times 10^{-3}$
N	0.0295±0.0862	−0.0021±0.1795	$2.9689\times 10^{-1}$	0.0689±0.0594	0.1196±0.1327	$2.2772\times 10^{-2}$
NI	−0.0050±0.0777	0.0054±0.1626	$3.6026\times 10^{-1}$	0.0602±0.0491	0.1258±0.1015	$5.7538\times 10^{-5}$
G	0.0035±0.0351	−0.0125±0.1377	$5.5856\times 10^{-1}$	0.0224±0.0271	0.1098±0.0824	$4.5398\times 10^{-9}$
**Sector**	${\bar{\bar{\Delta }}}_{S}$			$|{\bar{\bar{\Delta }}}_{S}|$		
	**Healthy**	**Glaucoma**	***p*-Value**	**Healthy**	**Glaucoma**	***p*-Value**
TS	0.0125±0.0853	−0.0055±0.3102	$6.4002\times 10^{-1}$	0.0642±0.0574	0.2357±0.1988	$2.4408\times 10^{-7}$
T	0.0108±0.0461	−0.0281±0.1472	$6.3324\times 10^{-2}$	0.0345±0.0324	0.1081±0.1025	$1.9310\times 10^{-5}$
TI	−0.0107±0.0752	−0.0356±0.3646	$5.9052\times 10^{-1}$	0.0591±0.0475	0.2817±0.2305	$6.6073\times 10^{-8}$
NS	−0.0391±0.0850	−0.0737±0.2296	$4.7228\times 10^{-1}$	0.0731±0.0581	0.1900±0.1463	$2.9558\times 10^{-5}$
N	0.0215±0.0637	0.0008±0.1630	$4.3602\times 10^{-1}$	0.0504±0.0443	0.1165±0.1127	$4.8909\times 10^{-4}$
NI	−0.0044±0.0844	0.0084±0.2420	$4.6913\times 10^{-1}$	0.0646±0.0543	0.1872±0.1511	$1.2741\times 10^{-6}$
G	0.0033±0.0346	−0.0162±0.1804	$5.4862\times 10^{-1}$	0.0214±0.0272	0.1428±0.1094	$9.0897\times 10^{-10}$

**Table 5 sensors-22-04842-t005:** Classification loss for observations not used for training ($w_{h}=1$, $w_{g}=1$).

ξ	δ	|δ|	Δ	|Δ|	$\bar{\Delta }$	$|\bar{\Delta }|$	$\bar{\bar{\Delta }}$	$|\bar{\bar{\Delta }}|$
15	0.2174	0.2222	0.1643	0.1498	0.1643	0.1932	0.1256	0.1546
10	0.1932	0.1932	0.1594	0.1498	0.1691	0.1739	0.1256	0.1498
5	0.1787	0.1643	0.1498	0.1546	0.1546	0.1449	**0.1159**	0.1449
3	0.1643	0.1401	0.1643	**0.1208**	0.1498	0.1304	0.1304	**0.1208**

**Table 6 sensors-22-04842-t006:** Performances of the best classification trees ($w_{h}=1$, $w_{g}=1$).

Tree Parameters	Accuracy	Sensitivity	Specificity	Precision
$\bar{\bar{\Delta }}$, ξ=5	0.8840	0.6595	0.9500	0.7948
|Δ|, ξ=3	0.8792	0.6170	0.9562	0.8055
$|\bar{\bar{\Delta }}|$, ξ=3	0.8792	0.5957	0.9625	0.8235

**Table 7 sensors-22-04842-t007:** Classification loss for observations not used for training ($w_{h}=1$, $w_{g}=1.5$).

ξ	δ	|δ|	Δ	|Δ|	$\bar{\Delta }$	$|\bar{\Delta }|$	$\bar{\bar{\Delta }}$	$|\bar{\bar{\Delta }}|$
15	0.2386	0.2516	0.1692	0.1844	0.2082	0.2234	0.1931	0.1909
10	0.2343	0.2408	0.1692	0.1844	0.2082	0.2148	0.1844	0.1866
5	0.2169	0.2082	**0.1432**	0.1996	0.1931	0.2104	0.1584	0.1822
3	0.1822	0.1497	0.1670	**0.1432**	0.1735	0.1627	0.1584	0.1562

**Table 8 sensors-22-04842-t008:** Performance of the best classification tree ($w_{h}=1$, $w_{g}=1.5$).

Tree Parameters	Accuracy	Sensitivity	Specificity	Precision
|Δ|, ξ=3	0.8792	0.6595	0.9437	0.7750

**Table 9 sensors-22-04842-t009:** Statistical characterization of the asymmetry metric $|\Delta_{S}{|}^{0.5}$ proposed for RNFL thickness. Mean values, standard deviations and *p*-values of this metric for the two subsets of healthy and glaucoma patients.

Sector	$|\Delta_{S}{|}^{0.5}$		
	Healthy	Glaucoma	*p*-Value
TS	0.1960±0.0931	0.3908±0.1741	$4.2798\times 10^{-9}$
T	0.1989±0.0929	0.3358±0.1856	$5.9084\times 10^{-5}$
TI	0.1870±0.0813	0.4393±0.2183	$5.6436\times 10^{-9}$
NS	0.2373±0.1019	0.3813±0.1739	$5.4446\times 10^{-5}$
N	0.2332±0.1145	0.3359±0.1623	$3.5263\times 10^{-4}$
NI	0.2190±0.1016	0.3842±0.1759	$3.5045\times 10^{-7}$
G	0.1254±0.0758	0.3469±0.1515	$2.4453\times 10^{-12}$

**Table 10 sensors-22-04842-t010:** Performance of the best classification tree ($w_{h}=1$, $w_{g}=1.5$).

Tree Parameters	Accuracy	Sensitivity	Specificity	Precision
${\left|\Delta \right|}^{0.5}$, ξ=3	0.8792	0.7021	0.9312	0.7500

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
