# Peer review of "Decision Trees for Glaucoma Screening Based on the Asymmetry of the Retinal Nerve Fiber Layer in Optical Coherence Tomography"

_sensors, 2022, doi:10.3390/s22134842_

Round 1

Reviewer 1 Report

Material and methods

-       The authors should provide a specific section with the criteria of inclusion and exclusion and the criteria used to consider the patients as “healthy” and “unhealthy/glaucoma”: which type of glaucoma was considered (primary/secondary/open angle/angle closure)? Also, the authors should provide some details on the demographic characteristics of the “287 individuals”

Discussion

-       The authors should add a “limitation” section to their discussion

-       The authors should provide insight about the future directions of this research 

Author Response

Please, see attached file.

Reviewer 2 Report

This manuscript by Berenguer-Vidal et al used decision trees based on the retinal nerve fiber layer (FNFL) thickness asymmetry from OCTs for the diagnosis of glaucoma, which is the second leading cause of blindness worldwide. The data are of high quality, it is suitable for publication with minor revision. I only have some minor issues.

1.     Please use “retinal nerve fiber layer” but not the abbreviation RNFL when it was first mentioned.

2.     Please change “it is the leading cause” to “ it is the second leading cause” (line 15) because cataract is the leading cause of blindness worldwide.

3.     I was wondering whether there is any RNFL structural differences between high intraocular pressure (IOP) associated glaucoma and normal tension glaucoma using the models they investigated. 

Thank you for the invitation.

Author Response

Please, see attached file.

Reviewer 3 Report

Glaucoma is an eye disease which damages the optic nerve head (ONH) and lead to irreversible blindness, worldwide. Optical coherence tomography (OCT) is an imaging modality used in the assessment of the glaucomatous damages, altering the ocular structures morphology. To discriminate the glaucoma from a healthy eye the evaluation of  the thickness of the retinal nerve fiber layer (RNFL) or the morphology of optic nerve

(ONH) is required. Indeed, asymmetry is considered as an early indicator of glaucoma. Other indicators of glaucoma are commonly: the difference in values of intraocular pressure, central corneal thickness, corneal hysteresis, neuroretinal rim width or RNFL thickness. The aim of this paper was to establish a set of asymmetry metrics based on the thickness of the RNFL of both eyes as a predictor of glaucoma.

Authors should be congratulated for their work. Manuscript is well-written, and tables and figures are clear. Despite the observations enlightened by the Authors, several points warrant a discussion:

1.    Authors should define the baseline characteristic of patients studied. Comorbidities as diabetes or chronic medications could negatively affect the eye function. To better apply the model to general population, a focus on patients’ baseline features is required;

2.    Authors should discuss other work in which RNFL and OCT were used, such as this recent prospective non-randomized matched-pair study (PMID: 33925202)

Author Response

Please, see attached file.

Round 2

Reviewer 1 Report

The authors addressed some of the comments to the best if their knowledge but some additional comments (to the original version of the manuscript) still need to please be addressed: before publication can be considered

-       The title is too long with multiple abbreviations: the authors should revise it and make it shorter 

-       Abstract:

o   The use of abbreviations should be revised in the entire manuscript: an abbreviation should be explained when used the first time, and then only the abbreviation should be used – for example line 3 “RNFL” should be explained; “optical coherence tomographies” at line 2 should be replaced by “optical coherence tomography (OCT)” and only the abbreviation “OCT” should then used

o   The authors should re-write the abstract in order to present it in a more structured way (aim/methods/results/conclusions), with or without the headers of the different sections

-       Introduction:

o   IOP is the major risk factors for glaucoma onset and progression, but it is well known that IOP is not the only risk factor, and there are types of glaucoma – such as normal tension glaucoma- that are not related to an increased IOP. Therefore, the authors should correspondingly re-word the statement “Glaucoma comprises ocular disorders with optic neuropathies related to a clinically characteristic intraocular pressure (IOP) [3].”

o   The definition of glaucoma should be more scientific and the authers should indicate the differences between primary and secondary types of glaucoma, and open-angle and angle-closure glaucoma at the beginning of the manuscript. They should then indicate specifically to which type of glaucoma they are referring to in their study (primary open-angle glaucoma?).

o   Line 53/entire manuscript: the use of abbreviations should be revised in the entire manuscript - an abbreviation should be explained when used the first time, and then only the abbreviation should be used – examples: ISNT rule, CDR, RDR should all be explained and not abbreviated the first time they are mentioned in the manuscript

Reviewer 3 Report

Authors answered all comments and suggestions 
